# Structural basis for cannabinoid-induced potentiation of alpha1-glycine receptors in lipid nanodiscs

Arvind Kumar [1], Kayla Kindig [1], Shanlin Rao [2], Afroditi-Maria Zaki [2], Sandip Basak[1], Mark S. P. Sansom [2], Philip C. Biggin [2] & Sudha Chakrapani [1,3,4] ✉

Nociception and motor coordination are critically governed by glycine receptor (GlyR) function at inhibitory synapses. Consequentially, GlyRs are attractive targets in the management of chronic pain and in the treatment of several neurological disorders. High-resolution mechanistic details of GlyR function and its modulation are just emerging. While it has been known that cannabinoids such as $\Delta^9$-tetrahydrocannabinol (THC), the principal psychoactive constituent in marijuana, potentiate GlyR in the therapeutically relevant concentration range, the molecular mechanism underlying this effect is still not understood. Here, we present Cryo-EM structures of full-length GlyR reconstituted into lipid nanodisc in complex with THC under varying concentrations of glycine. The GlyR-THC complexes are captured in multiple conformational states that reveal the basis for THC-mediated potentiation, manifested as different extents of opening at the level of the channel pore. Taken together, these structural findings, combined with molecular dynamics simulations and functional analysis, provide insights into the potential THC binding site and the allosteric coupling to the channel pore.

Inhibitory neurotransmission at the glycinergic synapses in the spinal dorsal horn is a key regulatory mechanism of nociceptive signaling for maintaining physiological pain sensitivity. Disruption of this pathway accentuates central pain signaling and leads to pathological pain states. Glycinergic neurotransmission across motor and sensory neurons in the brainstem and spinal cord is mediated by glycine receptors (GlyR) where, in addition to pain perception, they play a key role in respiration, motor coordination, and muscle tone[1–3]. Frontal lobe epilepsy, hyperekplexia, and chronic pain are some of the pathological conditions associated with dysfunctions of glycinergic neurotransmission[4–8]. Studies show that enhancing the spinal glycinergic tone through inhibition of glycine reuptake and positive allosteric modulation of GlyRs suppresses nociceptive signaling and alleviates chronic pain. GlyRs belong to the large superfamily of

pentameric ligand-gated ion channels (pLGIC) that also includes the nicotinic acetylcholine receptors (nAChR), serotonin-3-receptors (5-HT₃R), and γ-aminobutyric acid-A-receptors (GABAAR)[9–14] (https://doi.org/10.1038/nature13552), (https://doi.org/10.1038/s41586-018-0672-3).

Our mechanistic understanding of pLGIC function has tremendously increased in the light of high-resolution structural information of the channel in multiple functional states and in complex with several modulators. In recent years, several studies have elucidated the structure of homomeric GlyRs in the presence of agonists, partial agonists, and antagonists, which provide a high-resolution structural view of gating mechanisms in these channels[15–21]. GlyRs share a conserved architecture as other members of pLGIC, with each subunit consisting of an extracellular domain (ECD)- a twisted β-sheet of ten

[1]Department of Physiology and Biophysics, School of Medicine, Case Western Reserve University, Cleveland, OH, USA. [2]Department of Biochemistry, University of Oxford, Oxford, UK. [3]Cleveland Center for Membrane and Structural Biology, School of Medicine, Case Western Reserve University, Cleveland, OH, USA. [4]Department of Neuroscience, School of Medicine, Case Western Reserve University, Cleveland, OH, USA. ✉e-mail: Sudha.chakrapani@case.edu

strands, a transmembrane domain (TMD)- a four α-helical bundle, and an intracellular domain (ICD)- a primarily unstructured region between the third and fourth TM helices. Binding of glycine in the ECD triggers a conformational change across the three domains, leading to channel opening and then eventually desensitization, involving distinct gates within the TMD (Fig. 1a). The sequence of molecular events underlying channel gating is modulated by both endogenous and exogenous ligands including several drugs of abuse. Several of these modulators bind to conserved pockets within the TMD; however, the details of how these sites communicate to the channel pore are still unclear.

GlyR allosteric modulators are a large pool of chemically diverse small molecules that include neurosteroids, cannabinoids, general anesthetics, metal ions, and toxins. Apart from the AM-3607-allosteric modulation site discovered in the ECD of GlyR[22], the TMD of pLGICs represents a hotspot for modulation of channel function by various

modulators such as ivermectin, alcohols, neurosteroids, and general anesthetics[17,23–25]. Endocannabinoids and phytocannabinoids are being looked upon as excellent starting scaffolds for designing lead molecules with subunit-specific effects[26]. The *Cannabis sativa* plant, when used in adjunct with epilepsy medication, has been shown to be effective in reducing the frequency of seizures in some patients[27,28]. The main active ingredients of cannabis (or phytocannabinoids) are Δ⁹-Tetrahydrocannabinol (THC) and cannabidiol (CBD). While THC is responsible for producing the most subjective effects of cannabis, CBD lacks the psychoactivity of THC. Although the primary effects of cannabinoids are induced through activation of cannabinoid type 1 and type 2 receptors (CB1 and CB2), several cellular and behavioral effects have been shown to extend beyond these G-protein coupled receptors[29,30]. Recent studies have implicated GlyRs as an important direct target for cannabinoids (both endogenous endocannabinoids

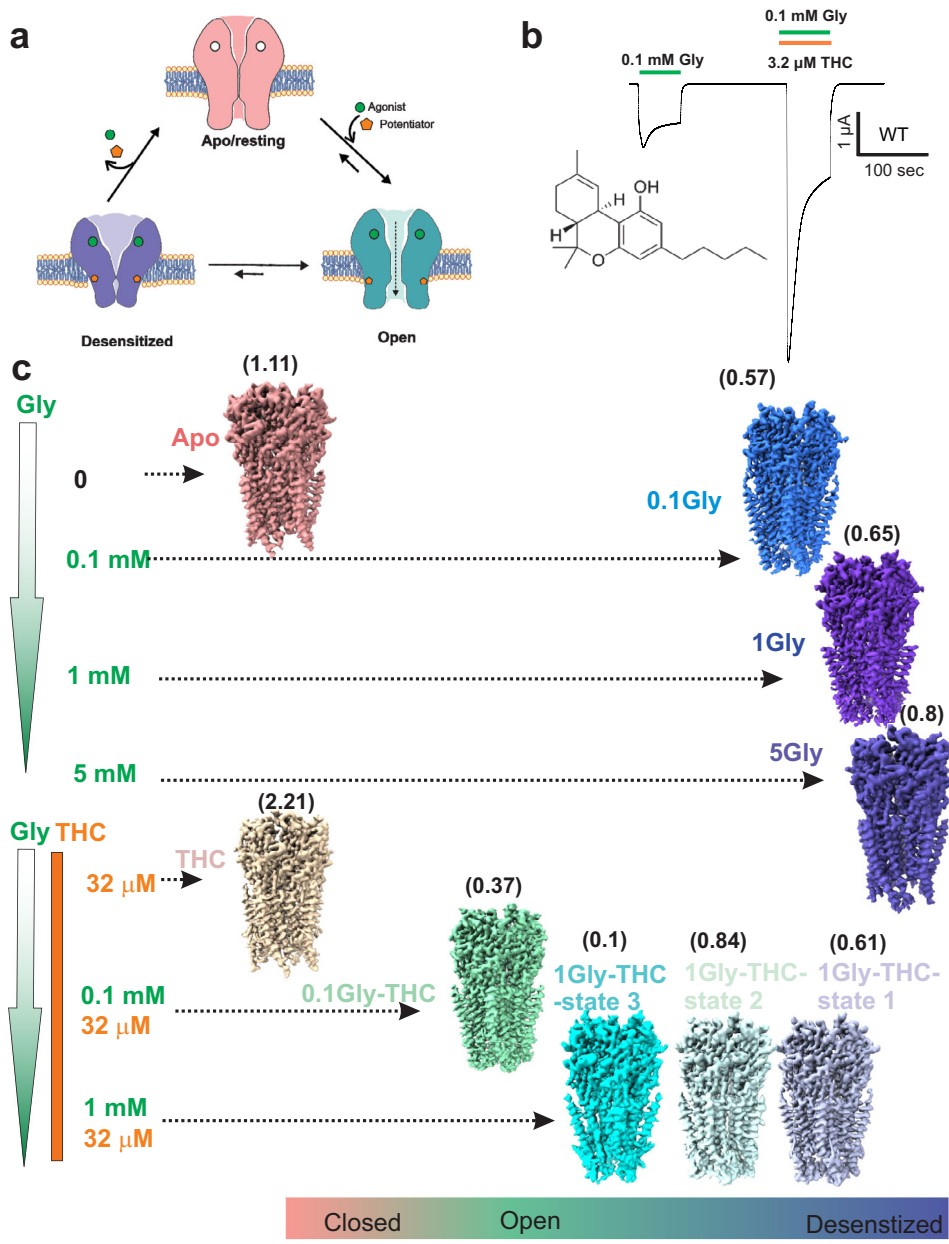

**Fig. 1 | Cryo-EM analysis of GlyR gating and THC modulation. a** Schematic representation of GlyR gating involving transitions from the Apo (resting) to glycine-bound, open and desensitized conformations. **b** TEVC recording of GlyR currents activated by 0.1 mM glycine, and with co-application of 3.2 μM Δ⁹-tetrahydrocannabinol (THC). Chemical structure of THC. **c** Cryo-EM 3D reconstructions of various GlyR conformations captured as a function of THC and Gly concentrations. A heuristic score is provided in parenthesis alongside each structure[95]. A cut-off value of Σd > 0.55 is used for predicting a non-conductive conformation. GlyR-Apo and GlyR-5gly were previously reported[19].

and exogenous phytocannabinoids) in the central and peripheral nervous systems in regulating neuromotor activity and pain[26,31,32]. In addition, rescue of cocaine-induced seizures by cannabinoids has been shown to be mediated through potentiating GlyR function (independent of CB receptors)[27]. It is noteworthy that cannabinoid-induced analgesia is absent in mice lacking GlyRα3 but not in mice lacking CB1 and CB2 receptors. THC has been shown to directly potentiate GlyR currents in clinically relevant concentration ranges[26,31,33] in both α1 and α3 GlyRs, the predominant α subunits in adult GlyRs (Fig. 1b).

While high-resolution structural information is now available for allosteric modulation of pLGIC by alcohols, neurosteroids, anesthetics, benzodiazepines, and ivermectin, little is known about cannabinoid modulation[19,23,24,34–36]. One of the main challenges to using cannabinoids for further drug development is our limited understanding of the drug-binding site, drug-receptor interactions, and mechanism by which allosteric effects are exerted across various functional domains of the channels. The first structural evidence for direct interaction of GlyR with THC came from high-resolution NMR studies of GlyRα1 TMD, where they identified a potential binding site for cannabinoids in a pocket lined by M3 and M4, in the vicinity of a serine residue in the middle of M3 (position Ser324 in human GlyRα1)[37]. Resonance peaks corresponding to this position showed notable differences in the absence and presence of THC. In NMR titration and NOESY experiments, intramolecular cross peaks were observed between cannabidiol and human GlyRα3 (equivalent position is Ser329), implicating a direct interaction of GlyRs with cannabinoids[38–40]. These results, combined with mutagenesis and functional analysis in cells and animal models, established that this serine position is critical for cannabinoid modulation. However, it remains unclear if the THC effect is exerted directly through binding to the serine sidechain or through an allosteric effect involving this position. Furthermore, the resonances of the rest of the TMD did not reveal any shifts, leaving the mechanism underlying THC potentiation still unknown.

Here we investigate the molecular details of THC-GlyR interactions, and probe the conformational rearrangements underlying allosteric potentiation by THC, using single particle cryo-electron microscopy (Cryo-EM) analysis of zebrafish homomeric GlyRα1 (referred to as GlyR) reconstituted in lipid-nanodiscs. The GlyR sample in complex with THC (32 μM) was imaged under various glycine concentrations. These structures, captured in multiple conformational states, reveal the molecular details of THC-binding pocket interactions and systematic conformational changes associated with THC binding. Molecular dynamics simulations were used to assess the conductance state of the various conformations and stability of the THC-binding pose. Combined with mutagenesis and electrophysiology, these findings elucidate the molecular basis for THC potentiation in GlyR.

## Results

### Cryo-EM structural titration to study the effect of THC on glycine-activated conformations

The codon-optimized full-length zebrafish glycine α1 receptor (GlyR) gene cloned into pFastBac1 plasmid (Supplementary Table 1), was used for baculovirus generation and protein expression in *Spodoptera frugiperda* (Sf9) cells[19]. The detergent solubilized and purified pentameric population of GlyR was reconstituted into asolectin nanodiscs with the E3D1 membrane scaffolding protein. In functional measurements, glycine acts as full agonist, activating GlyRs with an $EC_{50}$ of 0.1 mM and peak current response saturating at 1 mM[41]. THC does not evoke GlyR currents on its own up to micromolar concentrations, and the maximal potentiation is observed around 32 μM[37]. THC-potentiation of GlyR currents occurs at submaximal glycine responses, and the overall effect of THC disappears at glycine concentrations evoking maximal current (Supplementary Fig. 1). To capture the effect of THC on GlyRs at various concentration of glycine (0, 0.1, and 1 mM), the GlyR nanodisc samples were incubated with 32 μM THC and these

concentrations of glycine prior to vitrification on cryo-EM grids. For each of these glycine concentrations, the samples were also imaged without THC to allow direct comparison of the conformational states. The vitrified samples were imaged on Titan Krios 300 keV electron microscope and single-particle analysis was performed in RELION. The particles from the GlyR sample in complex with THC alone (GlyR-THC) belonged to one conformational state with the three-dimensional reconstruction at a nominal resolution of 3.09 Å (Supplementary Fig. 2). Analysis of GlyR in the presence 0.1 mM glycine without THC (GlyR-0.1gly) and with THC (GlyR-0.1gly-THC) also revealed one conformational state, each with resolutions of 2.60 Å and 2.80 Å, respectively (Supplementary Figs 3 and 4). Interestingly, while in the presence of 1 mM glycine alone, the particles sorted to one conformational state (GlyR-1gly) at 3.28 Å resolution (Supplementary Fig. 5), and in the presence of 1 mM glycine and 32 μM THC, the particles sorted to three classes: GlyR-1gly-State 1 (77, 813) at 2.91 Å, GlyR-1gly-State 2 (44, 653) at 3.57 Å, and GlyR-1gly-State 3 (18, 516) at 3.61 Å (Supplementary Figs. 6, 7, and 8). The 3D reconstructions in each state included density for most of the ECD and TMD residues, with the ICD being mostly unresolved. The density surrounding the TMD corresponds to the nanodisc belt consisting of helical segments of the membrane scaffolding protein and lipid bilayer within the enclosure. There are non-protein densities in the ECD that appear as an extension from Asn62 corresponding to N-glycans. In each case, the final reconstruction was used for model building and refinement. The final models did not include the region between Phe341-Lys394, which is a part of the M3-M4 loop in the ICD and lacks continuous density in the reconstructions.

We initially assessed each of these conformations using a simulation-free heuristic model that was derived using machine learning from MD simulations of nearly 200 ion channel structures[42]. Based on the plot of the local pore radii at each pore-lining residue against its side-chain hydrophobicity, the sum of shortest distances ($\Sigma d$) from positions falling in the low-likelihood region to the line is used as a heuristic score that allows a prediction for the likelihood of the conformation corresponding to a de-wetted and non-conductive state. A cut-off value of $\Sigma d > 0.55$ is used for predicting a non-conductive conformation. Based on the $\Sigma d$ values and the ligand conditions, the conformations were placed between conductive and non-conductive (Fig. 1c). While in the absence of glycine, both apo and the THC-complex had a large heuristic score (1.11 and 2.21, respectively) reflective of a non-conductive closed state; all glycine-bound conformations in the absence of THC had scores between 0.57-0.80, as previously reported for the desensitized conformation; co-complexes of glycine and THC had scores between 0.10-0.84, suggestive of mixed population of conductive and non-conductive states. Analysis of lipid-like densities in various glycine and glycine-THC reconstructions revealed distinct density in the reconstructions from THC-treated samples (Supplementary Fig. 9).

### GlyR-Apo and GlyR-THC structures

The analysis of the GlyR-THC structure revealed an overall conformation similar to the previously solved GlyR-Apo structure with an RMSD (pentamer) of 0.45 Å. Notably, the capping loop in the ECD (Loop C) adopts an outward conformation with no ligand density in the neurotransmitter binding pocket. The TM helices are oriented as in the GlyR-Apo structure with the M2 helices revealing multiple narrow constrictions along the ion permeation pathway at the Thr13′ (Thr289), Leu9′ (Leu285), and Pro-2′ (Pro274) positions (Supplementary Fig. 10). The pore dimensions at these positions are below the Born radius for the solvated chloride ion (2.26 Å)[43] and are therefore likely barriers to ion permeation. The Leu9′ sidechains in both the structures are oriented towards the pore axis and form a hydrophobic girdle, which performs the role of the activation gate of the channel. The Pro-2′ position is part of the charge-selectivity filter in anionic pLGICs,

comprising of a conserved stretch (Pro-2′, Ala-1′, and Arg0′)[44,45]. Additionally, the Pro-2′ position is the sole constriction point along the permeation pathway in anionic pLGICs in agonist-bound conformations, acting as the desensitization gate by limiting ion permeation in the desensitized state[19,35,46].

As previously shown, the hydrophobicity of the pore-lining residues may impede ion permeation even in pores that are not narrow enough for steric occlusion by causing local de-wetting and raising the energetic barrier for water and ion permeation[42]. To assess ion conductance for the GlyR-THC state, molecular dynamics simulations were carried out upon embedding the GlyR structure within a 1-palmitoyl-2-oleoyl-*sn*-glycero-3-phosphocholine lipid bilayer in the presence of 500 mM NaCl, and applying a transmembrane potential of +500 mV (*i.e.* positive at the cytoplasmic side). The elevated concentration and voltage were used to increase the likelihood of observing permeant ions in microsecond simulation times. As expected, and also as previously seen in the GlyR-Apo state, no ion permeation events were observed for GlyR-THC, even at this elevated potential and concentration, confirming that the GlyR-THC structure corresponds to a non-conductive, resting state-like conformation (Supplementary Fig. 10).

### GlyR-0.1gly and GlyR-0.1gly-THC

At 0.1 mM glycine, both in the absence and presence of THC, the 3D classification resulted in one conformational state in each case. In the 3D reconstructions for GlyR-0.1gly and GlyR-0.1gly-THC, a prominent density corresponding to glycine is visible in the neurotransmitter-binding pocket at the interface of the principal and complementary subunit (Fig. 2a, b). The glycine pocket is lined by Phe123, Phe183, Tyr226, Thr228, and Phe231 from the principal subunit and residues Phe68, Arg89, Leu141, Ser145, and Leu151 from the complementary subunits. At the level of the ECD, the two structures aligned with no

major differences in the side-chains of residues lining the glycine-binding pocket, except for a rotation of the Phe231 sidechain. In the GlyR-0.1gly-THC reconstruction, a lipid-like density is observed in an inter-subunit groove wedged between M3 and M4, and in close proximity to Ser320 (position Ser324 in human GlyRα1, previously shown to be critical for human GlyRα1 and GlyRα3 potentiation by cannabinoids[37]). We modeled the density as THC, oriented such that the methyl groups of the C10 monoterpene moiety face the hydrophobic groove lined by the Phe418 and Val421 sidechains on M4 and Phe317 on M3 (Fig. 2c). The opposite side of the monoterpene ring has the aromatic hydroxyl group that is oriented toward Ser320 (corresponding to Ser324 in human GlyRα1). This orientation is conducive to forming a hydrogen bond interaction between the aromatic hydroxyl group and Ser320 sidechain. THC binds in the vicinity of a classical lipid binding pocket of GlyR which is occupied by PIP₂ in a recent α1β3γ2 GABA_AR structure solved in nanodiscs[9]. The n-pentyl tail points downward and engages in a hydrophobic interaction with Ile410 and Val328 on M4 and M3, respectively (Fig. 2d). An alanine substitution at Ser320 decreased THC potentiation without any significant effect on the $EC_{50}$ for glycine[37]. Sequence alignment shows that while this position is a serine in α3 and α1 GlyRs, it is an alanine in GlyR α2 and β subunits (Supplementary Fig. 11). Substituting the alanine residue with a serine in GlyRα2 imparted THC sensitivity to the channel[37]. The other residues surrounding the THC density including Phe418 and Val421 are also conserved across various GlyR subtypes. We studied the effect of S320A mutation in zebrafish GlyR on THC potentiation by TEVC. Peak amplitude of glycine-evoked currents in the absence and co-application of THC was compared. Consistent with earlier studies, the S320A mutation significantly reduced the magnitude of potentiation (Fig. 2e). Co-application of THC with 0.1 mM glycine resulted

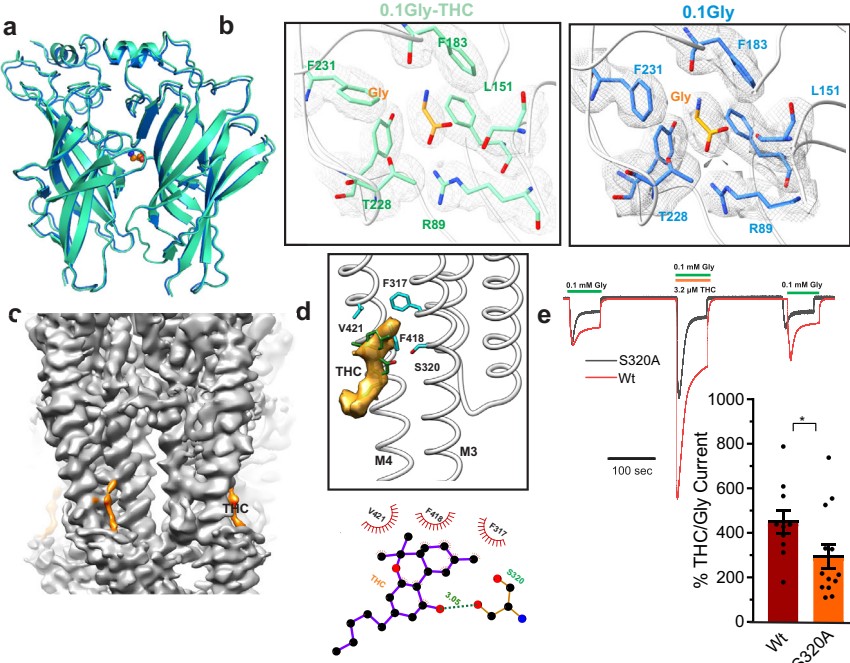

**Fig. 2 | THC modulation of GlyR gating. a** Alignment of GlyR-0.1gly and GlyR-0.1gly-THC structures. A side-view of the ECD-ECD subunit interface in the two structures. **b** Cryo-EM density segments for the neurotransmitter-binding site residues (gray) and the bound glycine molecule. **c** Cryo-EM map of GlyR-0.1gly-THC with THC density highlighted in orange. **d** THC molecule within the intrasubunit groove formed by the M3 and M4 helices *(top)*. LigPlot analysis[96] of THC interaction with the binding pocket. Residues within 4 Å distance are displayed in red *(bottom)*. **e** Continuous TEVC recordings of WT GlyR and S320A activated by 0.1 mM glycine,

and with co-application of 3.2 μM THC. Membrane potential was held at -60 mV. The S320A trace is scaled up x1.59 so that the first glycine-induced current amplitude matches WT GlyR, to emphasize the difference in degree of potentiation in response to THC. Percent potentiation is plotted as (peak of THC-glycine current/ peak glycine current) x 100 for WT (*n* = 10) and S320A (*n* = 13). Data are shown as mean ± s.e for (n) independent experiments. Electrophysiology experiments were performed on independent oocytes, from multiple different surgeries. Two-sided Mann–Whitney test *P = 0.0358. Source data are provided as a Source Data file.

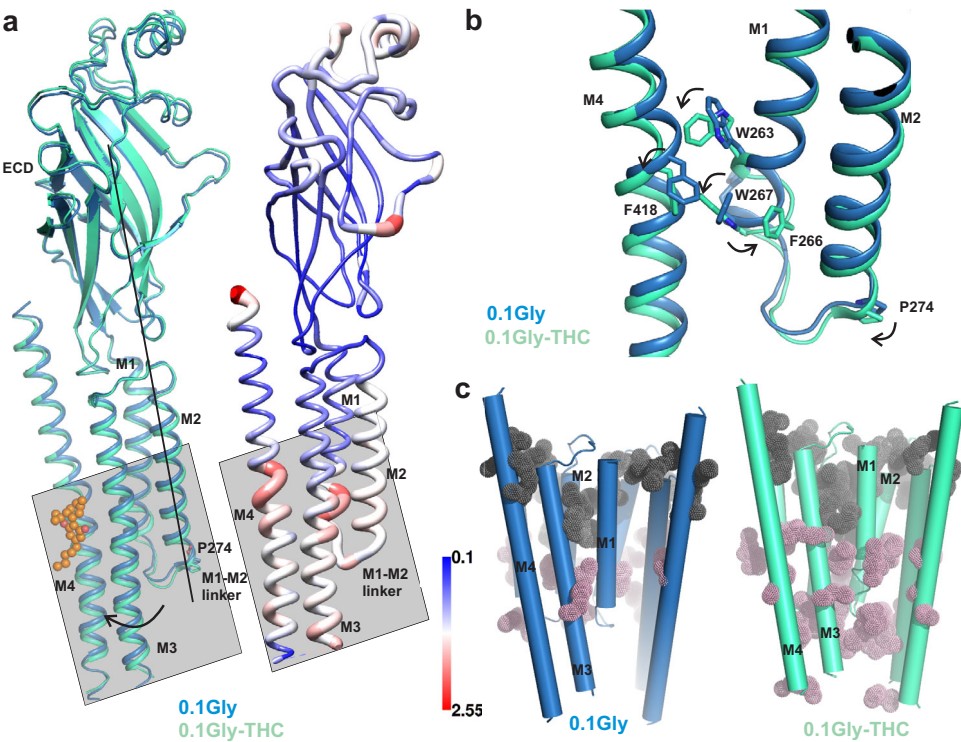

**Fig. 3 | THC-induced conformational changes within the TMD. a** A single-subunit alignment of GlyR-0.1gly and GlyR-0.1gly-THC structures highlighting the overall conformational changes associated with THC binding (left). Putty representation of pairwise root mean square deviation (RMSD) for the GlyR-0.1gly and GlyR-0.1gly-THC structures (right). The gray box points out regions that show most positional differences. **b** A close-up view of the alignment of GlyR-0.1gly and GlyR-0.1gly-THC structure showing rotameric rearrangement of residues in the vicinity of the THC binding site. **c** A comparison of the inter-subunit (gray dots) and intra-subunit (pink dots) cavities in GlyR-0.1gly (left) and GlyR-0.1gly-THC (right) predicted using F pocket algorithm[72]. The cavities predicted on the surface were removed for clarity.

in a potentiation of $449.5 \pm 53.7\%$ of the peak amplitude. For the S320A mutation, THC potentiation is reduced to $293.6 \pm 54.3\%$.

An alignment of GlyR-0.1gly and GlyR-0.1gly-THC structures reveals local structural changes in the mid-region of M4 and M3 helices involving residues Phe418, Ser320, and Phe317, and the intracellular ends of M1 and M2 (Fig. 3a). The aromatic side-chains at each of the positions Trp262, Trp267 and Phe266 undergo rotameric repositioning that is eventually propagated along the M1-M2 linker (Fig. 3b). The outward orientation of the M1-M2 linker moves Pro-2′ away from the pore axis, leading to the widening of the pore at the location of the desensitization gate. Interestingly, the concerted movement of M3 and M4 helices leads to changes in the intra and inter-subunit cavity volumes at the intracellular end of the channel (Fig. 3c). Notably, the intra- and inter-subunit cavities in the intracellular half of the GlyR-0.1gly-THC structure is larger in comparison to GlyR-0.1gly. A change in the volume and polarity of these cavities during gating has previously been suggested based on differential accessibilities of the residues lining the cavity during gating and allosteric modulation. Moreover, the accessibility of these residues increased in the glycine-bound conformations compared to the resting state[47,48].

An investigation of the M2 conformations in the two structures shows that the Leu9′ side chains are rotated away from the pore axis, and the pore at this position is opened to the same extent (Fig. 4a, e)[18,19,49]. A plot of pore radii along the ion permeation pathway reveals that the pore is wider than 3 Å throughout with exception of the Pro-2′ position (Fig. 4b, f). Interestingly, at Pro-2′, the pore is less constricted in the GlyR-0.1gly-THC when gauged from the pore radii and the Cα position (Fig. 4c, g). In the MD simulations of these structures, the pore is hydrated at the Leu9′ with chloride ions freely permeating this region (Fig. 4d, h). However, at the Pro-2′, the ions are somewhat impeded in GlyR-0.1gly as opposed to the frequent permeation

events observed in GlyR-0.1gly-THC. The GlyR-0.1gly-THC is selective for anions with no Na⁺ ions permeating through the Pro-2′ selectivity filter region. By counting the number of permeation events under these conditions of membrane potential and ionic concentration, we estimate a single-channel conductance (corrected to 150 mM NaCl) of $23 \pm 1$ pS for the GlyR-0.1gly-THC state. The conductance value is in close agreement to the previously estimated conductance (20 pS) for the open GlyR conformation in the presence of 5 mM glycine and trapped in the presence of a PTX blocker. The conductance estimate for GlyR-0.1gly was much lower at $9 \pm 2$ pS. Interestingly, in the simulation studies for the conformation captured in the GlyR-5gly structure, we observed only water permeation through the Pro-2′ with no chloride ions passing through the channel during 1 µs of total simulation time[19]. This finding highlights the difference in the energetic barrier for ion permeation at the Pro-2′ in the structures at 0.1 mM and 5 mM glycine concentrations.

## GlyR-1mM gly and GlyR-1mM gly-THC structures

The GlyR-1gly structure is similar to the GlyR-0.1gly and GlyR-5gly, in terms of twisting of the ECD, inward movement of Loop C, to expansion of the TMD. The M2 conformation of GlyR-1gly features an open pore at the level of Leu9′ and a narrow constriction at Pro-2′, suggesting an overall desensitized state (Supplementary Fig 12). In the presence of THC, the three conformational states had glycine occupancy in the binding site and the conformation of the ECD was similar. However, the structures differed in their M2 conformations. GlyR-1gly-State 1, which was the predominant population in particle distribution, was similar to GlyR-1gly in its overall structure including the M2 conformation and appears to be in a desensitized conformation. GlyR-1gly-State 2 and GlyR-1gly-State 3 had slightly wider pores at the level of M2 judging by the Cα position. We note that the resolution of these

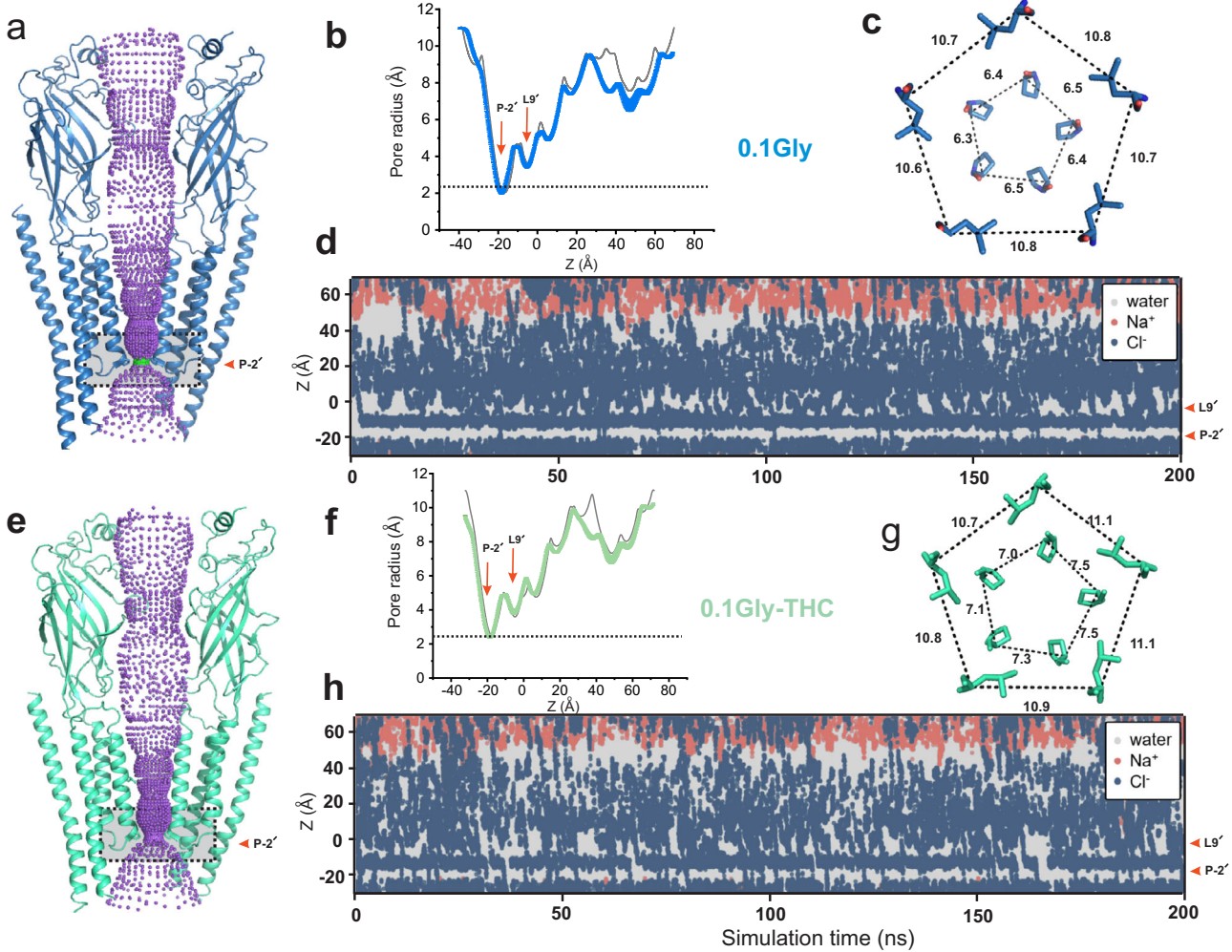

**Fig. 4 | Assessment of conductance state of GlyR conformations. a** Ion permeation pathway generated with HOLE for GlyR-0.1gly structure. For clarity, only two non-adjacent subunits are shown. Colors of the spheres represent the following pore radii: red <1.15 Å, green 1.8-2.3 Å and purple >2.3 Å. Residues lining various pore constrictions are shown as sticks. Gray box is shown to highlight the constriction at Pro-2′ position. **b** Mean pore radius and one-standard-deviation (from three independent 30 ns equilibrium simulations for the GlyR-0.1gly structure) plotted along the central pore axis *(colored lines)*. The final 20 ns of each 30 ns simulation trajectory was used to evaluate these profiles. Major constriction sites are indicated and the black dotted line denotes the radius of hydrated chloride ions. The gray line is the pore radius profile calculated from the cryo-EM structures. **c.** A view of Leu9′ and Pro-2′ positions from the extracellular end for the GlyR-0.1gly structure. Positions Leu9′ and Pro-2′ are shown in ball-and-stick representation and the corresponding distances between Cα are given in Å. **d** Simulation trajectories along the pore (z)-axis of water molecules and chloride ion coordinates within 5 Å of the channel axis inside the pore of GlyR-0.1gly structure, in the presence of a + 500 mV transmembrane potential difference (i.e., with the cytoplasmic side having a positive potential). One of five independent 200 ns replicates is shown. During these and the preceding simulations, positional restraints were placed on the protein backbone, in order to preserve the experimental conformational state while permitting rotameric flexibility in amino acid side chains. The energetic barriers due to the ring of Leu9′ and Pro-2′ are at z ~0 and −20 Å, respectively. **e** Ion permeation profile for the GlyR-0.1gly-THC structure. **f** Pore radius profiles and standard deviations averaged across three independent 30 ns equilibrium simulations for GlyR-0.1gly-THC structure performed using same conditions described above. **g** A view of Leu9′ and Pro-2′ from the extracellular end for the GlyR-0.1gly-THC structure. **h** Simulation trajectories along the pore of water molecules and chloride ion coordinates for the GlyR-0.1gly-THC structure.

conformations was considerably poorer (3.6 Å) than the other conformations reported here. In the MD simulations, all 1 mM glycine structures were stable with no major positional differences. Few permeation events were observed for GlyR-1gly, leading to a conductance estimate of 5 ± 2 pS. GlyR-1gly-State 1 and GlyR-1gly-State 2 had conductance of 3 ± 2 pS and 1 ± 1 pS, respectively, while GlyR-1gly-State 3 had a conductance of 10 ± 2 pS. Based on the conductance of various glycine-THC structures, it appears that while THC stabilizes an open conformation at 0.1 mM glycine, only a small fraction of the particles appears to be in a conductive conformation in the presence of 1 mM glycine. These findings are broadly consistent with the extent of potentiation observed at various glycine concentrations in electrophysiological recordings, which show that at 0.1 mM glycine, THC elicited over a four-fold increase (486.6 ± 72.5% potentiation) in peak

amplitude, whereas at 1 mM glycine only 30% increase in peak current was measured (133.7 ± 18.3% potentiation) (Supplementary Fig. 1).

### Validation of THC-binding pose
While a stronger density near Ser320 was seen in the 0.1gly-THC reconstruction, unambiguous modeling of the THC binding pose remained challenging even at 2.6-2.8 Å resolution ranges. In order to assess the stability of the THC binding pose and its interaction with the binding-site residues as seen in our structure, we carried out all-atom molecular dynamics simulation on GlyR-0.1gly-THC, which had the highest resolution of the GlyR-THC complex. These simulations were carried out either with Cα-restraints, to maintain the initial protein conformation, or in the absence of any positional restraints to allow a small amount of refinement of this initial pose. Further, unrestrained

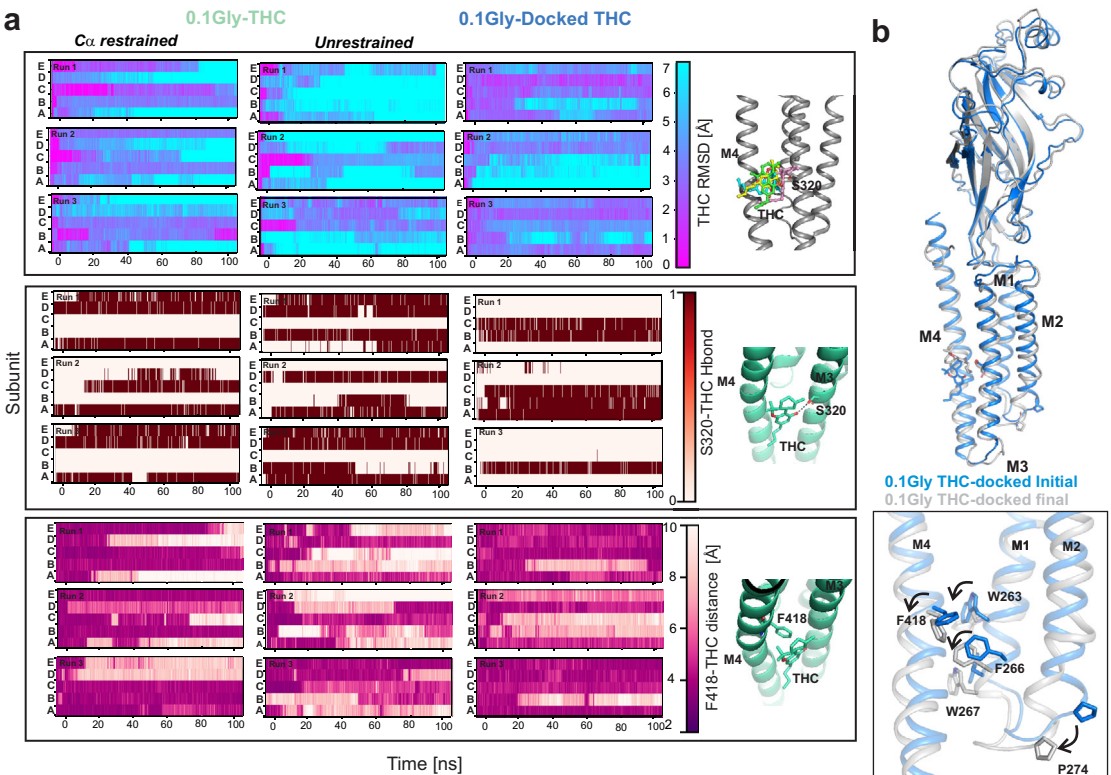

**Fig. 5 | Assessment of THC binding pose. a** The stability of the THC binding pose was assessed in the GlyR-0.1gly-THC structure during simulations with either Cα-restraints or under unrestrained conditions. The THC molecule was also docked on to the GlyR-0.1gly structure in the same binding pose as observed in the GlyR-0.1gly-THC structure. The time evolution of the RMSD of the heavy atoms of each THC molecule with respect to the starting conformation was computed for the three repeats for each simulated system (*top panel*). Prior to the calculation of each THC RMSD, a least-squares fitting was performed on the Cα atoms of the protein subunit where the respective THC was bound. Inset on the right shows the different THC orientations during unrestrained simulation. The number of hydrogen bonds formed between the oxygen or the hydroxyl group of THC and the Ser320 sidechain located on the M3 helix, for the three repeats for each simulated system. The criteria for considering hydrogen bond formation was that the O··O distance was below 3.5 Å and the H-O··O angle was below 30° (*middle panel*). The minimum distance between Phe418 sidechains and THC molecule during the three repeats for each simulated system (*bottom panel*). **b** Single subunit alignment of initial and final states of GlyR-0.1gly THC-docked structure highlighting overall conformational changes associated with THC binding during simulation.

simulations were carried out by docking THC molecules on to the GlyR-0.1gly structure (in the same pose as observed in the GlyR-0.1gly-THC structure) to assess local conformation changes within the TM helices upon THC binding. Both GlyR-0.1gly-THC and GlyR-0.1gly structures remained stable throughout the triplicate 100 ns simulation. The THC molecules in the Cα-restrained simulations stayed closer to the initial binding pose (in 2-3 subunits) as evidenced by the RMSD values (Fig. 5a, *top panel*). However, the THC RMSD was much higher in unrestrained simulations of GlyR-0.1gly-THC and docked- GlyR-0.1gly structures, as the THC molecule adopted multiple different orientations, particularly at the level of the pentyl chain. To evaluate the interaction of THC during simulation, hydrogen bond analysis between Ser320 and THC was carried out with the criteria that the O-O distance was below 3.5 Å and the H-O·O angle was below 30°. In the starting conformation, each of the five subunits has a hydrogen bond between the aromatic hydroxyl group of the THC and the sidechain hydroxyl group of Ser320. Some of these hydrogen bonds are maintained in all three repeats of Cα-restrained and unrestrained simulation runs (Fig. 5a, *middle panel*). This is noteworthy, considering the extent of flexibility displayed by the THC molecule within the binding pocket. Upon docking THC on to the GlyR-0.1gly structure, the THC molecule appears to establish the hydrogen bonding with Ser320 in some of the subunits, albeit to a lesser extent. Despite the fewer number of the hydrogen bonds, the THC molecule remains stable in the pocket. We note that this stability comes, at least in part, from hydrophobic interaction between THC and the aromatic ring of Phe418 on the M4 helix (Fig. 5a, *bottom panel*). For instance, the THC molecule remains

close to its starting conformation in chains D and E of docked GlyR-0.1gly simulation (Runs 1 and 3) even though there is no hydrogen bond with Ser320 in these chains. In both these chains, the proximity of THC rings with Phe418 is maintained throughout the runs. We then asked if THC induces conformational changes in the GlyR-0.1gly structure and how these changes relate to the conformation of GlyR-0.1gly-THC. Several aromatic residues in the vicinity of the THC binding pocket undergo rotameric reorientation (Fig. 5b). Particularly, Phe418, Trp263, Phe266, and Trp267 rotate and move in a direction similar to that seen in GlyR-0.1gly-THC. Quite remarkably, the M1-M2 linker, and hence the Pro274 residue are positioned away from the pore axis, leading to un-pinching of the pore at the desensitization gate (Supplementary Fig. 13).

To further probe the role of these residues on THC potentiation, we investigated the effect of mutations on THC potentiation for W263F, F266A, W267F, F418A, and P274A. Previous functional studies have shown that for W263F, P274A, S320A mutations, there is no change in dose response to glycine with EC50 values similar to WT[50–52]. On the other hand, F266A, W267F, F418A mutations have a right-shifted dose response compared to WT[50]. THC potentiation decreases with increasing glycine concentrations. Since none of the mutations increased glycine sensitivity, we measured macroscopic currents in response to 0.1 mM glycine in the absence and presence (co-application) of 3.2 μM THC (Fig. 6). Each of these mutations significantly decreased the THC effect on glycine-activated currents, and quite remarkably W263F, F418A, and P274A mutations almost completely remove potentiation. These findings are in agreement with the

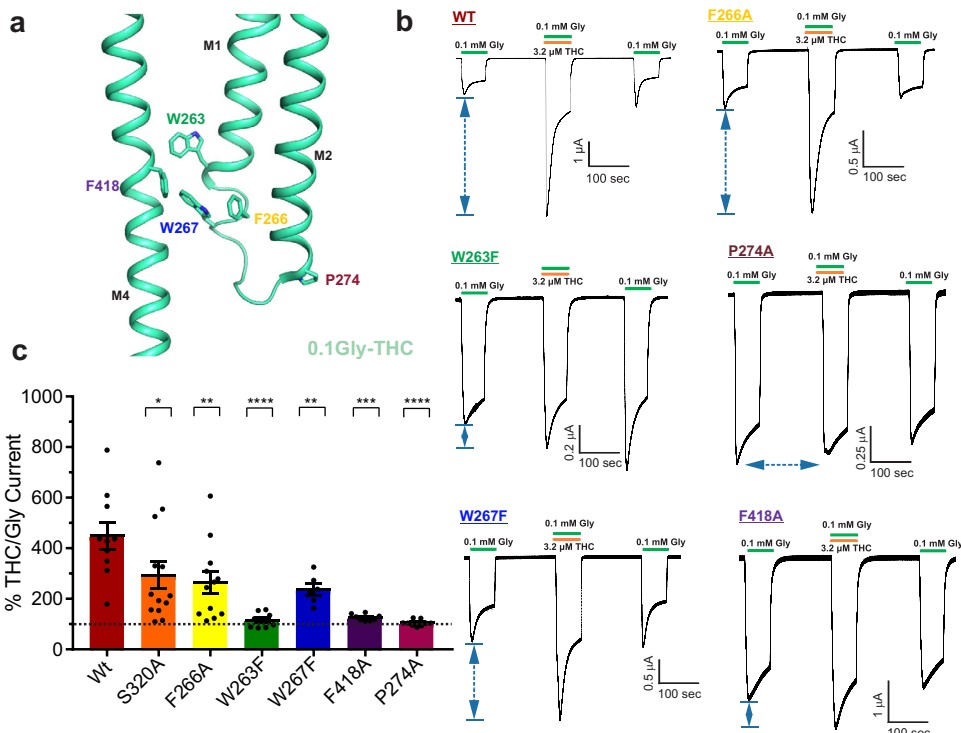

**Fig. 6 | Mutational analysis of THC-binding pocket. a** Residues lining the THC-binding pocket and along the putative coupling pathway. **b** Representative TEVC recordings of GlyR WT and mutants activated by 0.1 mM glycine, and with co-application of 3.2 µM THC. Membrane potential was held at −60 mV. **c** Percent potentiation is plotted as (peak of THC-glycine current / peak glycine current) x 100 for WT ($n = 10$), S320A ($n = 13$) W263F ($n = 9$), F266A ($n = 12$), W267F ($n = 6$), P274A ($n = 9$), F418A ($n = 8$). Data are shown as mean ± s.e for (n) independent experiments. Electrophysiology experiments were performed on independent oocytes, from multiple different surgeries. Two-sided Mann–Whitney test *$P = 0.0358$ (S320A), **$P = 0.009$ (F266A), **$P = 0.0075$ (W267F), ***$P = 0.0007$ (F418A), ****$P = 0.00005$ (W263F), ****$P = 0.00002$ (P274A). Source data are provided as a Source Data file.

predictions from MD simulation on the contribution of Phe418 sidechain in stabilizing the THC binding pose and further support the role of this aromatic network in coupling THC binding to the channel pore. In summary, the simulation highlights that while THC indeed forms a specific interaction with S320, it does not remain strongly bound to its original position during simulation. These findings are consistent with previous studies that demonstrated that chemical modification of THC by either removing the hydroxyl group (1-desoxy-THC) or removing the oxygen atom (5-desoxy-THC) did not alter the effect on GlyR potentiation. However, removing both polar groups (di-desoxy-THC) significantly reduced THC potentiation[37]. Enormous flexibility of the ligand within the binding pocket is characteristic of lipid-exposed binding sites. These sites can accommodate modulators of a range of molecular volumes[40]. A peripheral binding pocket at the lipid-protein interface also explains how membrane lipid constituents such as cholesterol affect THC efficacy[53].

A global comparison of glycine-THC structures with their corresponding glycine-only structures shows that the conformational differences are primarily at the plane of the THC binding pocket- the intracellular ends of M3 and M4, the M1-M2 linker, and the position of Pro-2′ (Supplementary Fig. 14). No structural change is observed in the ECD at the glycine-binding pocket or the interfacial loops associated with conformational coupling.

## Discussion

Chronic pain is a pathological phenomenon that poses a major health crisis affecting a significant portion of the adult population. Current pain management strategies, while better suited for acute pain, poorly serve patients with chronic pain, who in many cases are refractory to currently available analgesics. The GlyRs' therapeutic potential as a pain target has not been fully harnessed since compounds specific for GlyRs

are yet to be developed. In recent years, high-resolution structural studies of homomeric and heteromeric GlyRs have furthered our molecular understanding of gating and modulation in these channels[18–22]. Capitalizing on these advancements, in the present study, we explored the structural basis for THC potentiation of GlyR function. Here we present several structures of GlyR in complex with varying concentrations of glycine and THC that reveal GlyR-THC interactions and the range of conformational changes associated with positive allosteric modulation. Particularly, we show that THC binds in a lipid-exposed pocket at the interface of M3 and M4 helices, and is physically coupled, via a network of aromatic residues, to the M1-M2 linker and the Pro-2′ which constitute the desensitization machinery. Consistent with the previous studies, the binding site for THC was in the vicinity of S320. Structural changes at several positions further away from the THC binding pocket and into the intrasubunit cavity showed side-chain reorientation in the 0.1Gly-THC structure in comparison to the 0.1-Gly structure. Mutational perturbations at these positions decrease THC potentiation; notably in W263 and F418, the effect of THC was completely eliminated. The postulated allosteric pocket and the coupling pathway has implications beyond THC, and may extend to other cannabinoids including CBD, AG, and AEA[39,54]. Molecular dynamics simulations provide insights into the flexibility of the THC molecule within the pocket and its stable interaction with the conserved Ser320 position. Docking of THC molecule on the glycine-bound structure evokes structural changes observed in the cryo-EM structure.

It is conceivable that THC may adopt additional poses deeper within this cavity under physiological conditions. Along these lines, it was demonstrated that cholesterol depletion decreased THC potentiation in GlyR and implicated that this pocket may bind cholesterol to further stabilize THC[53,55]. While Cryo-EM sample preparations included cholesteryl hemisuccinate, it is likely that it may not capture the effects

of membrane cholesterol. In addition, it is to be noted that there are several lipid-like densities in the 0.1Gly-THC map that have not been modeled and could also be additional THC binding sites. Assignment of these densities is challenging and requires further investigation. Results from a recent high throughput in silico study suggest that, in addition to the binding site shown here, THC may utilize other pockets within the TMD, some of which have overlapping interactions (particularly with Ser320)[56].

Single-channel studies have shown that GlyRα1 exhibits five conductance states ranging from 18 pS to 90 pS[57]. The frequency of occurrence of these sub-conductance states is not dependent on the glycine concentration[58] and several residues along M2 including Gly278 (Gly2′) adjacent to Pro-2′ shift relative populations of various states[57]. Various glycine-bound structures, in the presence and absence of THC, reveal different extents of pore closures. Assessment of ion permeation events in molecular dynamics simulations provides estimates in the range 0-23 pS with the open conformation (0.1Gly-THC, and the previously determined 5 mMGly-PTX) having the highest single-channel conductance and the 5 mMGly state revealing no permeation events. The conductance estimates from our MD simulations are lower than the reported values measured experimentally. Therefore, it is challenging to annotate these structures to various subconductance states seen in single-channel recordings. Computational electrophysiology simulations of GlyRs (and pLGICs in general) are able to indicate relative but not absolute conductance values[59]. We believe that some of the differences may be due to the missing residues in the ICD that may exert an effect on the single-channel conductance through electrostatic mechanisms, but they may also reflect the limitations of the computational methodology.

THC has been previously shown to accelerate activation rates and decrease deactivation rate[26,37,40] with no significant effect reported on the macroscopic desensitization of the receptor[55]. It is intriguing that for glycine-bound conditions, THC stabilizes more open-like conformations when compared to structures with glycine alone. Detailed single-channel analysis of GlyR currents with THC is warranted to better understand the basis for this behavior. But on the other hand, there is also an intrinsic bias in single-particle Cryo-EM analysis where conclusions are drawn from particles with high-resolution signals, which in most cases represents only a small subset of the particles imaged. Perhaps future studies carried out with time-resolved Cryo-EM grid preparations, more extensive datasets, and further advanced processing tools may resolve many more of the sub-states that constitute the GlyR gating landscape including those of the heteromeric GlyRs.

In summary, the data presented in this study lay the groundwork to further investigate different lipidic modulators of GlyRs as well as other pLGIC members. While this study provides mechanistic details into potentiation of homomeric GlyRs, the challenges associated with uncovering mechanisms in heteromeric GlyRs and subunit-specific modulators still remain.

## Methods

### Electrophysiological recordings by two-electrode voltage-clamp (TEVC) in oocytes

The pCS2-a1 plasmid encoding zebrafish GlyRα1 for expression in *Xenopus laevis* was kindly provided by Prof. Robert Vandenberg, University of Sydney, Australia. (Supplementary Table 2). The mutations were introduced using standard site-directed mutagenesis protocol. To linearize the DNA, the plasmid was incubated with *Nde1* restriction enzyme at 37 °C for 2 h. The mRNA was synthesized from linearized DNA using the mMessage mMachine kit (Ambion) per instructions in the manufacturer's manual. The RNA was then purified with RNAeasy kit (Qiagen). About 0.2-30 ng of mRNA was injected into *X. laevis* oocytes (stages V–VI) and experiments were performed 1 day after injection. For control experiments to verify that no endogenous currents were

present, oocytes were injected with the same volume of water. Dr. W. F. Boron kindly provided oocytes used in this study. Female *X. laevis* were purchased from Nasco. We complied with all relevant ethical regulations for animal testing and research. Animal experimental procedures were approved by Institutional Animal Care and Use Committee (IACUC) of Case Western Reserve University. Oocytes were maintained at 18 °C in frog Ringer's solution (96 mM NaCl, 2 mM KCl, 1 mM MgCl$_2$, 1.8 mM CaCl$_2$, 20 mM HEPES) which was supplemented with 2.5 mM sodium pyruvate, 50 µg/mL gentamicin, and 100 µg/mL tetracycline, with pH adjusted to 7.5 and osmolarity to 195 mOsm.

TEVC experiments were performed on an Axon Instruments Axoclamp 900 A. Currents were sampled and digitized at 500 Hz with an Axon Digidata 1550B, and analyzed by Clampfit 10.7.0.3 (Molecular Devices). Oocytes were clamped at a holding potential of −60 mV and solutions were exchanged using a syringe pump perfusion system flowing at a rate of 6 ml/min. The electrophysiological solutions consisted of (in mM) 96 NaCl, 2 KCl, 1.8 CaCl$_2$, 1 MgCl$_2$, and 20 HEPES (pH 7.5, osmolarity adjusted to 195 mOsM). Glycine was purchased from Sigma-Aldrich and tetrahydrocannabinol (THC) was purchased from Cerilliant®. Current traces were plotted using Origin Version b9.5.0.193. Statistical analysis was performed using Graphpad Prism 7.04 (Graphpad Software). All data are reported as mean + s.e for (n) individual oocytes. No sample size calculation was made. All statistical tests were unpaired and two-sided. Normalization of traces was performed in Clampfit 10.7.0.3.

### Full-length GlyR cloning and transfection

The zebrafish GlyRα1 shares 92% amino acid similarity with the human GlyRα1. Codon-optimized zebrafish GlyRα1 (NCBI Reference Sequence: NP_571477) was purchased from GenScript (Supplementary Table 2). The sequence included the full-length GlyRα1 gene (referred to in the text as GlyR) followed by a thrombin sequence (LVPRGS) and a C-terminal octa-His tag. The gene was subcloned into the pFastBac1 vector for baculovirus generation and protein expression in S*podoptera frugiperda* (Sf9) cells[60]. The Sf9 cells (purchased from Invitrogen) were cultured in Sf-900™ II SFM medium (Gibco®) without antibiotics and incubated at 28 °C without CO$_2$ exchange. Transfection of sub-confluent cells was carried out with recombinant GlyR bacmid DNA using Cellfectin II transfection reagent (Invitrogen) according to manufacturer's instructions. Cell culture supernatants were collected at 5 days post-transfection and centrifuged at 1000 × g for 5 min to remove cell debris to obtain progeny 1 (P1) recombinant baculovirus. Sf9 cells were infected with P1 virus stock to produce P2 virus, which were then used for subsequent infection to produce P3 viruses and so on till the P6 generation. The P6 virus was used for recombinant protein production based on expression levels and eventually quality of the purified sample.

### GlyR expression, purification and nanodisc reconstitution

Approximately 500 ml of 2 × 10⁶ per ml Sf9 cells were infected with 1 ml o P6 recombinant viruses. After 48 h post-infection, the cells were harvested and centrifuged at 8000 × g for 20 min at 4 °C to separate the supernatant from the cell pellet. The cell pellet was resuspended in a dilution buffer (20 mM Tris-HCl, pH 7.5, 36.5 mM sucrose) supplemented with 1% protease inhibitor cocktail (Sigma-Aldrich). Cells were disrupted by sonication and non-lysed cells were removed by centrifugation at 3000 × g for 15 min in a bench-top centrifuge. The supernatant was subjected to ultracentrifugation at 167,000 g for 1 hr to separate the membrane fraction and the membrane pellet was frozen until further use. The membrane pellet was solubilized with 15 mM *n*-dodecyl-β-D-maltopyranoside (DDM, Anatrace) in a buffer containing 200 mM NaCl and 20 mM HEPES, pH 8.0 (buffer A), supplemented with 0.05 mg/ml soybean polar extract (asolectin, Avanti Polar Lipids) and 0.05% cholesteryl hemisuccinate (CHS, Avanti Polar Lipids) for 2 h at 4 °C. Non-solubilized materials was removed by ultracentrifugation (167,000 × g for 25 min). The supernatant

containing the solubilized protein was incubated with TALON resin pre-equilibrated with buffer A, 1 mM DDM, 0.05 mg/ml asolectin, and 0.05% CHS for 2 h at 4 °C. The beads were then washed with 10 column volumes of buffer A, 1 mM DDM, 0.05 mg/ml asolectin, 0.05% CHS, and 35 mM imidazole. The GlyR protein was eluted with buffer A, 1 mM DDM, 0.05 mg/ml asolectin, 0.05% CHS, and 250 mM imidazole. Eluted protein was concentrated and applied to a Superose 6 column (GE healthcare) equilibrated with buffer A and 1 mM DDM. Fractions containing the GlyR pentamers were collected and concentrated to 0.5 mg/ml using 50-kDa MWCO Millipore filters (Amicon).

The nanodisc reconstitution was carried out with the membrane scaffold protein (MSP1E3D1). MSP1E3D1 was expressed and purified with some modifications to the previously described protocols[61]. The MSP1E3D1 gene in pET28a (a gift from Stephen Sligar: Addgene plasmid # 20066) was expressed in *E. coli* BL21(DE3) cells grown in Terrific broth medium supplemented with kanamycin (25 $\mu$g mL$^{-1}$) and 0.2% glucose. The culture was grown at 37 °C with shaking to an $OD_{600}$ of ~1.0, and induced by with 1 mM IPTG for 4 h at 37 °C. The cell pellet was lysed in a buffer containing 100 mM NaCl, 20 mM Tris base, pH 7.4 and supplemented with 1 mM PMSF and complete EDTA-free protease inhibitor cocktail tablet (Roche). The lysate was centrifuged at 30,000 × $g$ for 30 min and the supernatant was bound to Ni-NTA, washed with four bed volumes of buffer (buffer B: 300 mM NaCl, 40 mM Tris, pH 8.0) and 1% Triton X-100, four bed volumes of buffer with 50 mM sodium cholate, four bed volumes of buffer B, four bed volumes of buffer B containing 20 mM imidazole, and eluted with buffer B containing 300 mM imidazole. The eluted MSP1E3D1 was passed through a desalting column equilibrated with 100 mM NaCl, 50 mM Tris, 0.5 mM EDTA, and pH 7.5, and the concentration was determined by absorbance at 280 nm (extinction coefficient = 29,910 M$^{-1}$ cm$^{-1}$). The purity was assessed by SDS-PAGE and size-exclusion chromatography.

For reconstitution, the soybean polar lipid extract (Avanti Lipids) was dried in a stream of nitrogen and equilibrated in buffer A with protein, DDM, and MSP1E3D1 such that final molar ratio of Protein: MSP: lipid: DDM was 1:3:360:1800. The mixture was incubated for 1 hr with gentle rotation. After 1-hr incubation, Bio-bead SM-2 (Bio-Rad Laboratories) were added and mixture was subjected to gentle rotation for 12 h at 4 °C. The reconstituted protein was applied to a Superose 6 column (GE healthcare) equilibrated with buffer A. Fractions containing protein reconstituted in nanodiscs were collected and concentrated to 0.2 mg/ml using 50-kDa MWCO Millipore filters (Amicon) for cryo-EM studies.

## Preparation of sample cryo-EM imaging and parameters for data acquisition

GlyR nanodisc sample from gel filtration (~0.15 mg/ml) was filtered and used for cryo-EM. For the Apo-THC condition, the sample was incubated with 32 $\mu$M THC for 30 min (GlyR-THC). For GlyR-gly and GlyR-gly/THC, the samples were incubated for 30 min with 0.1/1 mM glycine and a combination of 0.1/1 mM glycine and 32 $\mu$M THC, respectively. The sample was blotted twice with 3.5 $\mu$l sample each time, glow-discharged Cu 300 mesh Quantifoil R 1.2/1.3 grids (Quantifoil Micro Tools) coated with Graphene oxide (Sigma-Aldrich)[62] and the grids were plunge frozen into liquid ethane using a Vitrobot (FEI). The grids were imaged using a 300 keV FEI Titan Krios microscope equipped with a Gatan K3 direct electron detector camera and BioQuantum K3 Imaging Filter for all datasets. The parameters for data acquisition for the different sample conditions are as follows:

GlyR-THC: 5760 super-resolution movies containing 40 frames were collected at ×81,000 magnification (set on microscope) in a physical pixel size of 1.08 Å/pixel, dose per frame 1.25 e-/Å$^2$. Defocus values of the images ranged from −1.0 to −2.0 $\mu$m (input range setting for data collection) as per the automated imaging software Latitude (Gatan).

GlyR-0.1gly: 7844 movies with 70 frames/movie were collected with EPU (ThermoFisher Scientific) at 81,000× magnification (set on microscope) in super resolution mode with a physical pixel size of 1.1 Å/pixel, dose per frame 0.85 e-/Å$^2$. Defocus values ranged from −0.8 to −1.8 $\mu$m (input range setting for data collection).

GlyR-1gly: 5822 movies with 70 frames/movie were collected using EPU (ThermoFisher Scientific) at 81,000× magnification (set on microscope) in super-resolution mode with a physical pixel size of 1.1 Å/pixel, dose per frame 0.85 e-/Å$^2$. Defocus values ranged from −1.0 to −2.0 $\mu$m (input range setting for data collection).

GlyR-0.1gly-THC: For the 0.1 mM glycine and 32 $\mu$M THC, movies containing 50 frames were collected with EPU at ×130,000 magnification (set on microscope) in super-resolution mode with a physical pixel size of 0.68 Å/pixel, dose per frame 1.25 e-/Å$^2$. Defocus values of the images ranged from −1.0 to −2.0 $\mu$m (input range setting for data collection).

GlyR-1gly-THC: For the dataset with 1 mM glycine and 32 $\mu$M THC, movies containing 40 frames were collected at 81,000× magnification (set on microscope) in super-resolution mode with a physical pixel size of 1.1 Å/pixel, dose per frame 1.2 e-/Å$^2$. Defocus values of the images ranged from −1.0 to −2.0 $\mu$m (input range setting for data collection) as per the automated imaging software EPU.

**Image processing.** MotionCor2 (v1.2.3)[63] with a B-factor of 300 pixels$^2$ was used to correct beam-induced motion. Super-resolution images were binned (2 × 2) in Fourier space. All subsequent data processing was conducted in RELION[64] 3.0, 3.06, 3.1. The defocus values of the motion-corrected micrographs were estimated using Gctf software[65]. Similar strategies were used to process all the datasets and is described in detail for the GlyR-THC dataset.

**GlyR-THC dataset.** CTF-corrected micrographs were subjected to template-based autopicking in RELION 3.0. Templates generated using previously published GlyR-Apo dataset were used for autopicking. A total of 124,240 particles were auto-picked from 5760 micrographs and were subjected to 2D classification to remove suboptimal particles. 64,000 good particles were selected and subjected to initial round of 3D auto-refinement using 20 Å low-pass filtered map of GlyR-5mM Gly (EMD-20715). Multiple rounds of 2D classification and 3D classification without image alignment were done to remove low resolution and broken particles. All the particles corresponding to bottom views of the receptor were removed from subsequent processing to prevent misalignment of the particles in 3D auto-refinement. The nanodisc belt was subtracted to improve the angular accuracy of alignment and remove sub-optimal particles. Parameters for Bayesian polishing were obtained by training using 30% of total particles. Multiple rounds of Bayesian polishing were performed using estimated parameters and 3D classification was carried out to remove the particles with low resolution features. A final subset of 22,238 particles used for auto-refinement, with soft mask accounting for density of nanodisc, resulted in 3.09 Å map upon post-processing (Fourier shell coefficient FSC = 0.143 criterion). The B-factor estimation and map sharpening were performed in the post-processing step. Local resolutions (2.5 Å–6.5 Å) were estimated using the RESMAP software[66]. Additional map was generated using DeepEMhancer[67] using two half-maps generated during final rounds of autorefinement in RELION and was used for model building of the protein density.

**GlyR-0.1gly.** Overall strategy for data processing was as described for GlyR-THC dataset. A total of 147,734 particles were auto-picked from 7844 micrographs and were subjected to 2D classification to remove suboptimal particles. A final polished subset 89,791 particles used for auto-refinement, with soft mask accounting for density of nanodisc, resulted in 2.61 Å map upon post-processing (FSC = 0.143 criterion). The B-factor estimation and map sharpening were performed in the

post-processing step. Local resolutions (2.5 Å–6.5 Å) were estimated using the RESMAP software[66].

**GlyR-1gly.** Overall strategy for data processing was as described for GlyR-THC dataset. A total of 189,243 particles were auto-picked from 5822 micrographs and were subjected to 2D classification to remove suboptimal particles. A final polished subset 29,483 particles used for auto-refinement, with soft mask accounting for density of nanodisc, resulted in 3.28 Å map upon post-processing (FSC = 0.143 criterion). The B-factor estimation and map sharpening were performed in the post-processing step. Local resolutions (3 Å–7 Å) were estimated using the RESMAP software.

**GlyR-0.1gly-THC.** Overall strategy for data processing was as described for GlyR-THC dataset. A total of 190,380 particles were auto-picked from 5803 micrographs and were subjected to 2D classification to remove suboptimal particles. A final polished subset 29,664 particles used for auto-refinement, with soft mask accounting for density of nanodiscs, resulted in 2.84 Å map upon post-processing (FSC = 0.143 criterion). The B-factor estimation and map sharpening were performed in the post-processing step. Local resolutions (2.5 Å–6.5 Å) were estimated using the RESMAP software.

**GlyR-1gly-THC.** Overall strategy for data processing was as described for GlyR-THC dataset. A total of 323,908 particles were auto-picked from 4336 micrographs and were subjected to 2D classification to remove suboptimal particles. Multiple rounds of 3D classification, CTF refinement and polishing resulted in particles which were classified into 3 distinct classes. Each of these classes were separately polished and refined and resulted in 3 different reconstructions. Class 1, here referred as state 1 with 77,813 particles, resulted in 2.91 Å map upon post-processing (FSC = 0.143 criterion). Class 2, here referred as state 2 with 44,653 particles, resulted in 3.57 Å map upon post-processing (FSC = 0.143 criterion). Class 3, here referred as state 3 with 18,156 particles, resulted in 3.61 Å map upon post-processing (FSC = 0.143 criterion). The B-factor estimation and map sharpening were performed in the post-processing step. Local resolutions were estimated using the RESMAP software.

### GlyR model building

The 3D cryo-EM maps generated using DeepEMhancer[67] for each datasets used for model-building contained density for the entire ECD, TMD and a small region of the ICD. The previously solved structure of GlyR-Apo (PDB ID: 6UBS) was used as a starting model for each dataset. The residues were renumbered as per the sequence available in the UniProt database (O93430). The M3 and M4 helices were truncated based on the quality of density in the respective maps. After initial model building in COOT, each of the structures were refined against their corresponding EM-derived maps using the phenix.real_space_refinement tool from the PHENIX software package[68,69], using rigid body, local grid, NCS, and gradient minimization. The individual models were then subjected to additional rounds of manual model fitting and refinement. The refinement statistics, the final model to map cross-correlation evaluated using phenix module mtriage, and the stereo-chemical properties of the models as evaluated by Molprobity[70] are detailed in the Supplementary Tables 3 and 4. The pore profile was calculated using the HOLE program[71]. The cavities were analyzed using Fpocket algorithm[72]. Figures were prepared using PyMOL v.2.0.4 (Schrödinger, LLC), (CorelDraw v.20.1.0.708).

### Molecular dynamics simulations

For conductance measurements, each structure was embedded[1] into phospholipid (POPC, 1-palmitoyl-2-oleoyl-*sn*-glycero-3-phosphocholine) bilayer membranes in five independent replicates. The protein-membrane systems were solvated by 500 mM of aqueous NaCl. MD simulations were performed in GROMACS 2018[73], using the TIP4P/2005 water model[74] and either the OPLS all-atom protein force field with united-atom lipids[75] or the AMBER 99SB-ILDN force field[76] with (all-atom) Slipids[77–79]. The temperature and pressure were maintained at 310 K and 1 bar during simulations, with the velocity-rescaling thermostat and a semi-isotropic Parrinello and Rahman barostat[80]. Their coupling constants $\tau_T$ and $\tau_P$ were 0.1 ps and 1 ps, respectively. Long-range electrostatic interactions were treated using the smooth particle mesh Ewald method[81] with a real-space cut-off of 1 nm, a Fourier spacing of 0.12 nm, and charge interpolation through fourth-order B-splines. The time integration step size was 2 fs. Covalent bonds were constrained through the LINCS algorithm[82]. To preserve the overall protein conformational state whilst allowing for local rotameric flexibility of amino acid side chains, harmonic restraints were placed on protein backbone atoms, with a force constant of 1000 kJ mol$^{-1}$ nm$^{-2}$. A transmembrane potential of +500 mV was applied via a uniform external electric field, with positive potential on the cytoplasmic side. Conductance values were calculated from the number of Cl$^-$ ions traversing each channel structure per unit time and averaged between five 200 ns replicates.

### Stability of THC binding-pose

Coordinates for GlyR-0.1gly-THC and GlyR-0.1gly were used as the starting points for the molecular dynamics simulations. In order for the PDB structures to be used in unrestrained molecular dynamics simulations, it was necessary to add a linker to connect the M3 and M4 helices. Therefore, Modeller version9.25[83] was employed to replace the QHKELAATAEEMR motif with a small (AGT) linker. Ten models were generated and the DOPE and QMean scores[84] were used for evaluation. After selecting the best model, the in-built loop-refinement tool of Modeller was used to generate another ten models, and the final model was selected again based on DOPE and QMean scores. The ligands glycine and THC were parametrized with the second-generation General Amber force field (GAFF2)[85]. The Amber 99SB-ILDN force field[76] was adopted for the GlyR in conjunction with the TIP4P2005 water model[74]. The POPC lipids were represented with the Slipids force field[77–79].

The GlyR/Gly/THC complex was first oriented in a conformation vertical to the xy-plane and was placed in the center of a pre-equilibrated POPC lipid bilayer containing 512 lipids, after it was ensured that the TM helices of the protein overlap with the hydrophobic region of the lipid membrane. The protein/ligands complex heavy atoms were position-restrained with a harmonic potential and a strong force constant equal to 10$^5$ kJ mol$^{-1}$ nm$^{-2}$, and the inflateGRO tool[79,86] was executed to rescale the lipid atom coordinates by a factor of four. A series of 25 iterations of shrinking and energy minimizing the system was performed until the POPC area per lipid was 70 Å$^2$, a little above the equilibrium area per lipid of 65.8 Å$^2$[87]. The system was subsequently solvated and any water molecules inserted in the lipid region were deleted. NaCl was added to neutralize the system and induce the physiological ion concentration of 0.15 M.

Molecular dynamics simulations were performed using the GROMACS software package (version 2020)[88]. Initially, the system was energy minimized with the steepest descent algorithm until the maximum force did not exceed 100 kJ mol$^{-1}$ nm$^{-1}$ or until it converged to machine precision. A short equilibration run in the NVT ensemble was carried out for 5 ns and the temperature was stabilized at 310 K with the Nosé-Hoover thermostat[89,90] and a coupling constant of 0.5 ps. An additional equilibration step was performed subsequently in the NPT ensemble for 10 ns, with the same thermostat and a time constant of 0.5 ps, while the target pressure of 1 bar was reached using the Parrinello-Rahman barostat[80,90], and a semi-isotropic coupling with a time constant of 5 ps. Periodic boundary conditions were applied in all three dimensions. The hydrogen-containing bonds were constrained using the LINCS algorithm[91]. Neighbor searching, Lennard-Jones

interactions and real-space Coulomb interactions were cut off at 10 Å. The particle-mesh Ewald summation[92] was adopted for the treatment of the reciprocal-space electrostatic interactions with a cubic interpolation and a grid spacing of 1.2 Å. The time step was 2 fs and the leap-frog algorithm was selected for the integration of the equations of motion. Coordinates were saved every 20 ps. Three repeats of unrestrained 100 ns-long MD runs were subsequently performed for each system, amounting to a total simulation time of 600 ns. For the GlyR-0.1gly-THC system, in addition to the unrestrained simulations, a set of three MD runs was also performed with the protein Cα atoms positionally restrained with a force constant equal to 100 kJ mol$^{-1}$ nm$^{-2}$. The aim was to maintain the experimentally resolved protein structure while assessing the stability of THC in the binding site. The total simulation time amounted to 900 ns. In-built GROMACS tools and the MDAnalysis package[93] were used for trajectory analysis and PyMol for visualization[94].

### Reporting summary

Further information on research design is available in the Nature Research Reporting Summary linked to this article.

## Data availability

The data that support this study are available from the corresponding authors upon reasonable request. The cryo-EM maps have been deposited in the Electron Microscopy Data Bank (EMDB) under accession codes EMD-23700 (GlyR-THC), EMD-23702 (GlyR-0.1gly-THC), EMD-23701 (GlyR-0.1gly), EMD-23704 (GlyR-1gly-THC-state1), EMD-23705 (GlyR-1gly-THC-state2), EMD-23706 (GlyR-1gly-THC-state3), and EMD-23703 (GlyR-1gly). Coordinates have been deposited in the RCSB Protein Data Bank (PDB) under accession code 7M6M (GlyR-0.1gly-THC), 7M6O (GlyR-0.1gly-THC), 7M6N (GlyR-0.1gly), 7M6Q (GlyR-1gly-THC-state1), 7M6R (GlyR-1gly-THC-state2), 7M6S (GlyR-1gly-THC-state3), and 7M6P (GlyR-1gly). Source data underlying Figs. 2e, 6c, and Supplementary Fig. 1d are provided as a Source Data file. Source data are provided with this paper.

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

## Acknowledgements

We are grateful to the Cryo-Electron Microscopy Core at the CWRU School of Medicine and Dr. Kunpeng Li for the access to the sample preparation and Cryo-EM instrumentation. We acknowledge the use of instruments at the National Cryo-Electron Microscopy Facility at the NCI and Stanford-SLAC Cryo-Electron Microscopy Facility. We are very grateful to the members of the Chakrapani lab for critical reading and comments on the manuscript. This research was, in part, supported by the National Cancer Institute's National Cryo-EM Facility at the Frederick National Laboratory for Cancer Research. Research in MSPS's group was supported by grants from Wellcome (208361/Z/17/Z), BBSRC (BB/N000145/1) and EPSRC (EP/R004722/1). Computing was supported via the Advanced Research Computing facility, Oxford, the ARCHER UK National Supercomputing Service and JADE (EP/T022205/1) granted via the High-End Computing Consortium for Biomolecular Simulation, (HECBioSim - http://www.hecbiosim.ac.uk), supported by EPSRC (EP/R029407/1). AMZ and PCB were supported by BBSRC (BB/S001247/1). This work was supported by the National Institutes of Health grants R01GM131216, R35GM134896, and Cryo-EM supplements: 3R01GM108921-03S1, R01GM108921-5S1, 3R01GM131216-1S1 to S.C. and the AHA postdoctoral Fellowship to A.K. (20POST35210394) and S.B. (17POST33671152).

## Author contributions

A.K. and S.C. conceived the project and designed the experimental procedures. A.K. purified the protein, and optimized the cryo-EM sample preparation and performed grid screening. A.K carried out cryo-EM data collection, analysis, model building, and refinement with inputs from S.B. K.K. carried out mutagenesis, performed two-electrode voltage-clamp recordings and analyzed the electro-physiology data. S.R. performed the MD simulations, under the supervision of M.S. A.M.Z. performed the MD simulations under the supervision of P.C.B. S.C. supervised the execution of the experiments, data analysis, and interpretation. A.K. and S.C. drafted the paper with contributions from K.K., A.M.Z., S.B., S.R., P.C.B., and M.S. All authors reviewed the final paper.

## Competing interests

The authors declare no competing interests.
