## [Peer Review File · Nature Communications]

Structural Basis for Cannabinoid-induced Potentiation of alpha1-Glycine Receptor in Lipid NanodiscReviewers' Comments:

Reviewer #1:

Remarks to the Author:

Review of cryoEM data for "Structural Basis for Cannabinoid-induced Potentiation of alpha1-Glycine Receptors in Lipid Nanodiscs" by Kumar and co-workers.

The authors report a number of SPA Glycine receptor structures in the presence of THC. The previous Reviewer #2 had concerns about certain aspects of the cryoEM data analysis and placement of the THC molecule. I have very similar concerns to this reviewer. However, my concerns are not so much in the rescaling of pixel size (or collecting data at different magnifications and comparing structures under such a regime). My major concerns are with the placement of THC and the map interpretation. The authors in their rebuttal state:

The putative-binding pocket and the interactions with S320 side-chain are in agreement with previous NMR studies (Xiong Nat. Chem Biol 2011; Wells J. Med Chem 2015).

However Xiong et al. shows S296 is near the binding site? This numbering is different than the supplied PDB. Is amino-acid numbering wrong in PDB? Also Xiong et al. do not conclusively show that S296 is the binding site, only that THC addition slightly changes the chemical shift values (also according to Figure 3 of their manuscript many other residues exhibit chemical shift change). So, this paper may not be the strongest evidence for the binding site. Also Wells et al. report the results of a virtual screen with no experimental evidence to support S296 being the 'only' or conclusively the binding site for THC. These references do not add a lot of weight the best place to model THC.

I also tend to agree with Reviewer #2, that if you perform MD experiments with your putative binding site to show that this the binding site is proving things in a circular manner, as I agree that the THC binding site is not unambiguous.

The main reason I am slightly concerned with the current THC binding site is that there are many other densities around lipid binding pockets in maps that have not been modelled and the resolution around these regions is poor and all look like 'lipids'. For example: In the map GlyR_32uM_THC_1mM_Gly_State1.mrc there are lots of other 'lipid' like density which has not been modelled and the THC could fit equally well. (i.e. near L323, near F330, near W267, near Y303 etc....). These sites here could equally fit cholesterol, the terpene structures of THC, and some could fit multiple POPC/POPE like lipids especially with two POPC lipids next to each other in the current THC site in the 32uM_THC_1mM_Gly_State1 map.

There could be some other methods the authors could employ to better resolve these lipid binding pockets. The first could be to use a symmetry expansion approach and combine all the monomers from their C5 reconstructions and do a single monomer C1 refinement and mask out large regions, focussing on local Euler angle searches near the THC binding site. Secondly no 3D-variability analysis was performed. It is not clear if these extra densities are due to larger flexibility/dynamics. This may be informative also when comparing the Glycine only structure with the THC structures.

I think to compare each structure the 'exact' same mask width should be used in all final consensus refinements such that the proposed THC binding site had the same masking regime for all final 3D-refinements. This would ensure that this region was had its alignments considered in a similar manner. Some more minor comments:

1. The local resolution plots in the supplemental figures are misleading. They all are from 2ang to 8ang, but there is no FSC data in any of their maps to justify calculating local resolution below 2.6 ang (with most being above 3 ang). To my eyes (without the halfmaps on hand) it appears that the resolution around the THC binding sites is much poorer than indicated in the local resolution maps, with resolution looking more like 4-5 ang in these regions.
2. Model corresponding to 0p1G-THC-0p665_postprocessed.mrc has many side chain errors where the sidechains have been incorrectly modelled and are not in the observed density (i.e. F320, R295, K434, R437, K406 etc... carefully check whole structure). THC site is very much not convincing in this map.
3. DeepEMhancer was used, this also adds a level of concern to the reviewer as the authors are trying to report the THC binding site and an Deep Neural Network which hasn't been trained on such maps/data may add a level of convolution and misrepresent the noisy lipid binding sites.
4. I note that the authors opted quite a high defocus range for data collection (1-2 micro defocus).

Our group has found that a collecting much closer to focus and a smaller defocus range can better preserve high resolution data and helps with driving membrane protein structures to higher resolution. Just a tip for next time.

Matthew Belousoff, Monash Institute of Pharmaceutical Sciences, Australia

We thank the Reviewer for taking the time to provide a critical assessment of our manuscript. We appreciate the inputs and suggestions to improve the analysis. We have addressed the concerns to the best of our ability, and we believe that the manuscript has significantly improved in response to the feedback from this Reviewer and the previous round of reviews.

Review of cryoEM data for “Structural Basis for Cannabinoid-induced Potentiation of alpha1-Glycine Receptors in Lipid Nanodiscs” by Kumar and co-workers. The authors report a number of SPA Glycine receptor structures in the presence of THC. The previous Reviewer #2 had concerns about certain aspects of the cryoEM data analysis and placement of the THC molecule. I have very similar concerns to this reviewer. However, my concerns are not so much in the rescaling of pixel size (or collecting data at different magnifications and comparing structures under such a regime). My major concerns are with the placement of THC and the map interpretation.

The authors in their rebuttal state: The putative-binding pocket and the interactions with S320 side-chain are in agreement with previous NMR studies (Xiong Nat. Chem Biol 2011; Wells J. Med Chem 2015). However Xiong et al. shows S296 is near the binding site? This numbering is different than the supplied PDB. Is amino-acid numbering wrong in PDB? Also Xiong et al. do not conclusively show that S296 is the binding site, only that THC addition slightly changes the chemical shift values (also according to Figure 3 of their manuscript many other residues exhibit chemical shift change). So, this paper may not be the strongest evidence for the binding site. Also Wells et al. report the results of a virtual screen with no experimental evidence to support S296 being the ‘only’ or conclusively the binding site for THC. These references do not add a lot of weight the best place to model THC. I also tend to agree with Reviewer #2, that if you perform MD experiments with your putative binding site to show that this the binding site is proving things in a circular manner, as I agree that the THC binding site is not unambiguous.

Regarding the numbering of the serine position, our apologies for the confusion. The S296 (in the Xiong paper) is the same position as S320 (in our pdb). Some differences in the numbering are due to human vs zebrafish and whether the numbering includes the signal sequence, but there are also differences in numbering for human GlyRs between papers. We have clarified in the manuscript by changing the numbers to as they appear in UniProt (human GlyR α 1- S324, human GlyR α 3- S329, Zebrafish GlyR α 1- S320)

Previous mutagenesis studies in GlyRa1 and the finding that GlyRa2 (which has an alanine at this position) is insensitive to cannabinoids, underscore the importance of this position in cannabinoid-induced potentiation. Substituting the alanine residue with a serine in GlyRa2 imparted THC sensitivity to the channel. In the Xiong et al 2011 Nat. Chem Biol paper, they show changes in chemical shift for human GlyR α 1- S324 with THC. In a follow-up study with human GlyR α 3-TMD, they carry out NMR titration and NOESY experiments with other cannabinoids including cannabidiol, and show the appearance of intermolecular NOESY cross peaks between CBD and GlyR α 3 S329 TMD. While the latter is a stronger indicator of direct binding, we agree that these experiments are not conclusive about direct interaction with the serine residue. We state in the text “These results, combined with mutagenesis and functional analysis in cells and animal models, established that this serine position is critical for cannabinoid modulation. However, it

remains unclear if the THC effect is exerted directly through binding to the serine sidechain or through an allosteric effect involving this position.”

Our rationale for THC placement was first guided by these NMR and mutagenesis studies, but as we probed further, the lipid-like density at this site was stronger in GlyR-THC complexes, and the conformational differences between structures with and without THC were centered around this site. Additionally, our mutational analysis of positions W263F, F266A, W267F, P274A, S320A and F418A (that show the conformational differences) reveals that each of these mutations impacts THC potentiation, and quite remarkably W263F, F418A and P274A completely eliminate the THC effect.

We agree with the reviewer that our assignment of the binding pose to this density is not unambiguous (as we also state in the text). To assess the stability of the modeled pose, we had carried out all-atom, unrestrained molecular dynamics simulation on the GlyR-0.1gly-THC structure. We also carried out simulations of docking THC in the Cryo-EM pose on to the GlyR-0.1gly structure to monitor the ensuing structural changes. To these studies, we now added (triplicate) α -restrained simulations to see if the THC fluctuates in the binding pocket when the protein backbone is restrained to that of the Cryo-EM structure (Revised Figure 5). When compared to the unrestrained simulation, the THC is relatively more stable in its starting conformation in the α -restrained simulations.

In addition, in the simulations with docking THC on to the GlyR-0.1gly structure, the THC molecule remains stable in the pocket despite the fewer number of H-bonds with Ser320. We note that this stability comes, at least in part, from hydrophobic interaction between THC and the aromatic ring of Phe418 on the M4 helix (Fig. 5A, bottom panel). In both these chains the proximity of THC rings with Phe418 is maintained throughout the runs. Quite remarkably the F418A mutation almost completely removes potentiation. These findings are in agreement with the predictions from MD simulation on the contribution of Phe418 sidechain in stabilizing the THC binding pose and further support the role of this aromatic network in coupling THC binding to the channel pore.

The main reason I am slightly concerned with the current THC binding site is that there are many other densities around lipid binding pockets in maps that have not been modelled and the resolution around these regions is poor and all look like 'lipids'. For example: In the map GlyR_32uM_THC_1mM_Gly_State1.mrc there are lots of other 'lipid' like density which has not been modelled and the THC could fit equally well. (i.e. near L323, near F330, near W267, near Y303 etc...). These sites here could equally fit cholesterol, the terpene structures of THC, and some could fit multiple POPC/POPE like lipids especially with two POPC lipids next to each other in the current THC site in the 32uM_THC_1mM_Gly_State1 map.

We recognize that unambiguous modeling of the THC binding pose will be a challenge even if we improve the resolution to better than 2.8 Å since the proposed binding site lies at the lipid-protein interface and, as the Reviewer correctly points out, there are several additional lipid-like densities that could be potential binding sites for THC. Along these lines, a recent simulation study predicts that THC may exert its potentiating effects on GlyRs by binding to multiple pockets. We have discussed this ambiguity in the text in the discussion section (described later).

There could be some other methods the authors could employ to better resolve these lipid binding pockets. The first could be to use a symmetry expansion approach and combine all the monomers from their C5 reconstructions and do a single monomer C1 refinement and mask out large regions, focussing on local Euler angle searches near the THC binding site. Secondly no 3D-variability analysis was performed. It is not clear if these extra densities are due to larger flexibility/dynamics. This may be informative also when comparing the Glycine only structure with the THC structures. I think to compare each structure the 'exact' same mask width should be used in all final consensus refinements such that the proposed THC binding site had the same masking regime for all final 3D-refinements. This would ensure that this region was had its alignments considered in a similar manner.

We thank the Reviewer for the suggestions to improve the density for THC and other lipid-like molecules. We have explored all of these suggestions.

First, we evaluated the effect of different masks on the overall quality of density for lipids and THC by comparing 0.1Gly-THC map generated using the original mask and with the 0.1Gly mask in Relion 4.0 beta. The 0.1Gly-THC mask is slightly tighter than the 0.1Gly mask. The molecular weight inside the protein mask is 513111 for 0.1Gly-THC mask and 633072 for 0.1Gly mask. With the new map the signal for the THC was slightly better, but the overall resolution dropped from 2.87 to 3.01 Å.

Secondly, to further improve THC signal, we carried out symmetry expansion of the final polished 0.1Gly-THC particles, followed by local refinement. The inspection of THC density at individual subunits shows slightly better defined shape.

Third, the symmetry expanded 0.1Gly-THC particles were subjected to signal subtraction with single subunit mask, and 3D classification without image alignment. The 3D classification was done at various regularization parameters $T = 10, 20, 40, 80$. 3D classification with different T values did not result in distinct classes with or without THC density. In most cases at higher T -

value ~95% particles were found in one single class and remaining 4 classes usually contained remaining particles which were at too poor resolution to derive any conclusion.

We also tried 3DVA in CryoSparc to see if we can resolve different THC orientation. We could see THC density ranging from strong to weaker (and lacking continuity) in the various frames but couldn't resolve different binding modes or movements within the binding site.

Reviewer Figure 2. Symmetry expansion of final polished 0.1Gly-THC particles followed local refinement performed in RELION 4.0 beta (top panel). The THC density is highlighted by a circle at each of the five individual subunits. The map is contoured at σ level of 0.004.

29664 final particles were Symmetry expanded (148320 particles)
followed by particle subtraction (mask on single subunit)

Reviewer Figure 3. Workflow for symmetry expansion of 0.1Gly-THC particles was followed by signal subtraction to mask single subunit density, and 3D classification without image alignment. Analysis performed in Relion 4.0 beta.

Since we couldn't significantly improve the map quality for THC, we have highlighted the ambiguity of THC binding poses in the results and discussion section.

(Page 13) While a stronger density near Ser320 was seen in the 0.1Gly-THC reconstruction, unambiguous modeling of the THC binding pose remained challenging even at 2.6-2.8 Å resolution ranges.

(Page 15) In summary, the simulation highlights that while THC indeed forms a specific interaction with S320, it does not remain strongly bound to its original position during simulation. These findings are consistent with previous studies that demonstrated that chemical modification of THC by either removing the hydroxyl group (1-desoxy-THC) or removing the oxygen atom (5-desoxy-THC) did not alter the effect on GlyR potentiation. However, removing both polar groups (di-desoxy-THC) significantly reduced THC potentiation. Enormous flexibility of the ligand within the binding pocket is characteristic of lipid-exposed binding sites. These sites can accommodate modulators of a range of molecular volumes. A peripheral binding pocket at the lipid-protein interface also explains how membrane lipid constituents such as cholesterol affect THC efficacy.

(page 17) It is conceivable that THC may adopt additional poses deeper within this cavity under physiological conditions. Along these lines, it was demonstrated that cholesterol depletion decreased THC potentiation in GlyR and implicated that this pocket may bind cholesterol to further stabilize THC. While Cryo-EM sample preparations included cholesteryl hemisuccinate, it is likely that it may not capture the effects of membrane cholesterol. In addition, it is to be noted that there are several lipid-like densities in the 0.1Gly-THC map that have not been modeled and could also be additional THC binding sites. Assignment of these densities is challenging and requires further investigation. Results from a recent high throughput in silico study suggest that, in addition to the binding site shown here, THC may utilize other pockets within the TMD, some of which have overlapping interactions (particularly with Ser320).

Some more minor comments:

1. The local resolution plots in the supplemental figures are misleading. They all are from 2ang to 8ang, but there is no FSC data in any of their maps to justify calculating local resolution below 2.6 ang (with most being above 3 ang). To my eyes (without the halfmaps on hand) it appears that the resolution around the THC binding sites is much poorer than indicated in the local resolution maps, with resolution looking more like 4-5 ang in these regions.

We have updated Supplemental figures 2-8 with new resmaps. The scales shown are 2.5-6.5 Å for structures in the resolution range 2.6-3.1 Å and 3.0-7.0 Å for structures in the resolution range the 3.2-3.6 Å. The 0.1Gly-THC local resolution map is zoomed in to show the THC density and the neighboring region (Supplemental Figure 4, panel C)

2. Model corresponding to 0p1G-THC-0p665_postprocessed.mrc has many side chain errors where the sidechains have been incorrectly modelled and are not in the observed density (i.e. F320, R295, K434, R437, K406 etc... carefully check whole structure). THC site is very much not convincing in this map.

We have removed the 0p1G-THC-0p665_postprocessed.mrc and model. This was previously included to show the THC-induced conformational differences were not affected by map scaling.

3. DeepEMhancer was used, this also adds a level of concern to the reviewer as the authors are trying to report the THC binding site and an Deep Neural Network which hasn't been trained on such maps/data may add a level of convolution and misrepresent the noisy lipid binding sites.

We want to clarify that the DeepEMhancer maps were only used for aiding the modeling of the protein regions. We have now mentioned this in the methods.

4. I note that the authors opted quite a high defocus range for data collection (1-2 micro defocus). Our group has found that a collecting much closer to focus and a smaller defocus range can better preserve high resolution data and helps with driving membrane protein structuresa to higher resolution. Just a tip for next time.

Thank you! We are trying for this for recent data sets.

Reviewers' Comments:

Reviewer #1:

Remarks to the Author:

I think the authors have done an admirable effort to address the criticisms of this manuscript. I am satisfied that they have made every effort to make the manuscript clearer and address the concerns raised.

While the placement of the THC ligand still is ambiguous the text has been adjusted accordingly and the authors note in their rebuttal that while THC may have an allosteric effect and may not bind tightly to their proposed binding site, but the weight of their experiments and preceding NMR and mutagenesis experiments suggests that their placement is the best approximation.

I would recommend the manuscript for publication and would like the reviewers comments and rebuttals to also be published in case the readers would want some more details.

Regards,

Matthew Belousoff, Monash Institute of Pharmaceutical Sciences

Reviewer #2:

Remarks to the Author:

The manuscript reports molecular insights into conformational changes in homo-pentameric alpha-1 glycine receptors ($\alpha 1$ GlyR) mediated by the positive allosteric modulator $\Delta 9$ -tetrahydrocannabinol (THC). GlyRs are potential therapeutic targets for chronic pain and other neurological disorders. Cannabinoids, including THC, can produce analgesic effects through potentiating GlyRs. The potential clinical benefits of using GlyRs and cannabinoids, however, have not been yet fully capitalized due to limited knowledge on the interplays between GlyRs and cannabinoids. Thus, structural studies like the ones reported in the current manuscript are timely important for discovery and development of novel analgesics.

In the reported studies, the cryo-EM structures along with the molecular dynamics (MD) simulations provide novel molecular details regarding how THC positively modulates $\alpha 1$ GlyR function at a given glycine concentration. The research approach is robust. The overall quality of data (cryo-EM, MD simulations and functional measurements) and presentation are excellent, even though the THC map is not as strong as desired. The relatively weak cryoEM map for THC triggered some concerns in the previous review. However, it is not uncommon in cryo-EM structural determinations that, even with a good overall structure resolution, certain regions of a protein could not be well resolved or not resolved at all due to intrinsic dynamics of the regions. Results from the MD simulations (a large range of THC RMSDs, Fig. 5A) are in line with a weak THC map in the cryo-EM data, suggesting dynamical binding of THC. Is it reasonable to assign the THC binding site based on a weak map? Fortunately, the authors do not have to solely rely on the map to determine the THC site. As the authors pointed out, previous NMR and mutagenesis/functional experiments from other groups have shown that THC or CBD binds to the site with the lining residue S296 (equivalent to S320 in the current manuscript) in the transmembrane domain of GlyR (Xiong et al. Nat Chem Biol., 2011; Xiong et al. J Exp Med, 2012). The binding site was unambiguously confirmed by GlyR NOESY cross peaks with CBD (occurred only if CBD is in close contact with GlyR) and supported by GlyR chemical shift perturbation by THC. The authors also did their own mutagenesis/functional experiments to support the assigned THC binding site. All of these contribute to the assignment of the THC binding site instead of merely depending on the weak cryo-EM intensity.

This reviewer is satisfied with the authors' rebuttals to the previous review. They have adequately

addressed the concerns regarding the weak THC map and offered a balanced view in their revised manuscript. The new findings from the reported studies are valuable not only to understand the cannabinoid modulation of GlyRs but also to help discover new analgesics acting through the THC site. Thus, it is likely the reported studies will have a strong impact on both the research and pharmaceutical fields.